# Effect of High-Intensity Strength and Endurance Training in the Form of Small Circuits on Changes in Lipid Levels in Men Aged 35–40 Years

**DOI:** 10.3390/jcm11175146

**Published:** 2022-08-31

**Authors:** Tadeusz Ambroży, Łukasz Rydzik, Zbigniew Obmiński, Michał Spieszny, Antoni Szczepanik, Dorota Ambroży, Joanna Basiaga-Pasternak, Jakub Spieszny, Marta Niewczas, Jarosław Jaszczur-Nowicki

**Affiliations:** 1Institute of Sports Sciences, University of Physical Education, 31-571 Krakow, Poland; 2Department of Endocrinology, Institute of Sport—National Research Institute, 01-982 Warsaw, Poland; 3Department of General, Oncological, Gastrointestinal and Transplantation Surgery, Medical College, Jagiellonian University, 30-688 Krakow, Poland; 4Faculty of Physical Education and Sport, Institute of Social Sciences, University of Physical Education in Krakow, 31-571 Krakow, Poland; 5College of Medical Sciences, Institute of Physical Culture Studies, University of Rzeszow, 35-958 Rzeszow, Poland; 6Department of Tourism, Recreation and Ecology, University of Warmia and Mazury in Olsztyn, 10-719 Olsztyn, Poland

**Keywords:** lipid metabolism, cholesterol levels, training period, blood

## Abstract

Background: Blood lipid profiles consist of total cholesterol (TC) and its fractions, high-density lipoprotein cholesterol (HDL), low-density lipoprotein cholesterol (LDL), non-high-density lipoprotein cholesterol (non-HDL), and triglycerides (TG). For several decades, studies have examined the effects of various factors on lipid status and its association with the risk of developing arteriosclerosis and cardiovascular disease. The beneficial effects of increased physical activity on cardiovascular health have been demonstrated by appropriate modulation of lipid profiles. For individuals with low physical activity, the literature recommends engaging in various forms of training that can improve physical fitness and resting lipid status. The aim of the study was to examine whether a specific original training program improves lipid profiles to the levels recommended for the male population. Methods: The study involved two equal (*n* = 15) groups of men (experimental and control groups, aged 35–40 years). The experimental group performed 60-min training sessions for 8 weeks (3 times a week) including a set of strength and endurance exercises. Before and after the training program, blood was drawn from both groups for serum determination of TC, HDL, LDL non-HDL, and TG, and a battery of four field physical performance tests was administered. Results: Statistically significant decreases (TC by 19.3%, TG by 23.7%, LDL by 15%), a non-significant decrease (10% for non-HDL), and no change for HDL were found in the experimental group. Control group showed a statistically significant decrease, by 7.4% for TC. The results confirm the effectiveness of the proposed training in improving health indices. Conclusions: The 8-week training program met the health-related fitness paradigm recommended for physical activity in men aged 35–40 years. After the completion of the program, all the participants expressed their satisfaction from participating in a health-promoting experiment.

## 1. Introduction

Increasing human life expectancy, especially in developed countries, makes it necessary to monitor the quality of life of the middle-aged population and to prepare these people for progressive aging. This is not only a challenge for medicine but also leads to many problems to be addressed in the field of physical culture and public health. A group of the most common age-related diseases observed in modern society has been defined as diseases of affluence, which include cardiovascular diseases, hypertension, type 2 diabetes, cancers, autoimmune diseases, and the often-concomitant obesity [1]. The main risk factors for diseases of affluence, prevailing not only in highly developed countries, are improper diets and lifestyles, including low physical activity [2,3]. The greatest threat to health due to diseases of affluence are cardiovascular diseases, often leading to premature deaths. An early warning sign of the risk of such diseases is hypertension, often undiagnosed but also occurring in people before the age of 40. For this reason, many cardiologists recommend increasing physical activity to prevent hypertension [4]. Recently, the physical and mental health of the general population may have further deteriorated due to the significant reduction in physical activity and social contacts caused by the COVID-19 pandemic. As a result of the months-long lockdown, many individuals have reported decreased sleep quality, psychophysical well-being, and increased body weight [5,6,7].

The above-mentioned health problems increase with age, and therefore healthy lifestyle strategies, including increased participation in physical activity, should be developed and taught in the early stages of life. The measurable result of physical activity is physical fitness. In the 1990s, the concept of the determination of optimal health-related fitness (H-RF) levels emerged. It encompasses broadly defined cardiorespiratory functions, relative slenderness, muscle strength, endurance, and flexibility, recognized as essential factors in achieving a higher quality of life and counteracting health problems. Such a set of indices allows for the determination of the age-related range of a person’s ability to undertake everyday activities vigorously and energetically and is related to a low risk of the early development of diseases and a decline in vitality with age, [8,9] and consequently a decreasing proportion of high-intensity physical activity [10].

In Poland, a periodic and comprehensive assessment of health status is obligatory for economically active people, which allows for the estimation of the individual risk of developing certain diseases of affluence. An important element in such examinations is the determination of a biochemical marker of lipid metabolism in blood, i.e., cholesterol levels. It has been known for a long time that lipid metabolism disorders, measured by the proportion of cholesterol fractions in blood serum, are a significant risk factor for atherosclerosis, coronary heart disease, myocardial infarction, stroke, and atherosclerosis of the lower extremities [11,12,13]. In the case of high TC levels, a lipid profile is examined, with additional determination for HDL, LDL, and TG. For some time now, lipid profile studies have evaluated an additional component of cholesterol defined as non-HDL-C, which is the sum of LDL fraction and very-low-density lipoprotein (VLDL), which is atherogenic triglyceride-rich apolipoprotein B. The fraction defined as non-HDL has a higher diagnostic value than LDL in assessing the risk of arteriosclerosis and, consequently, the development of cardiovascular disease [14,15,16].

In routine diagnostic tests, LDL levels are not determined but calculated from the Friedewald formula developed in 1972 [17]. The validity of this formula, in which LDL is a linear function of three variables (TC, HDL, and TG), has been analyzed over the last few decades. To verify the reliability of the calculation results, the results obtained using the Friedewald method were compared with those of direct analytical measurements. Comparisons revealed little discrepancy between the data obtained from the two procedures, but as TG levels increased, the data from the calculation became increasingly underestimated compared to the analytical results [18]. It was assumed that analytical and calculated results for LDL do not match when the TG levels were >4.52 mmol/L (400 mg/dL), whereas in healthy subjects, at TG levels of ≤4.52, the calculated LDL levels were on average only 13% lower than the analytically determined value [19]. Recently, the formula for calculating LDL has been modified, in which LDL is expressed as a quadratic function (second-order polynomial) of TG [20]. The new formula has slightly better accuracy if there is concomitant hypertriglyceridemia, but the results are similar to those obtained using the classic Friedewald formula at normal TG ranges.

The close association between an unfavorable lipid profile and the risk of sudden cardiac death (SCD) has prompted researchers to search for extrinsic and individual factors responsible for lipid levels. In addition to higher body mass index and low physical activity, advanced age has also been shown to be responsible for SCD [21]. With age, endothelial function deteriorates, which in turn reduces arterial elasticity and nitric oxide (NO) bioavailability. This age-related phenomenon has also been observed in former competitive athletes [22]. Total cholesterol levels that are optimal for health depending on gender and age have been established. In men aged 35–45, concentrations of 200–230 mg/dL are the most beneficial, while higher or lower values increase the risk of mortality. Interestingly, too low concentrations of less than 140 mg/dL are particularly life-threatening [23]. Regardless of external factors, LDL levels and the effectiveness of interventions aimed to reduce them are genetically determined. A cardiovascular beneficial genotype that is responsible for low LDL levels [24] has been identified, while two respective gene variants are responsible for the different post-exercise directions of HDL, LDL changes [25]. Genetic factors may therefore explain interindividual differences in responses of cholesterol fraction to the same lifestyle.

Another factor that disrupts the lipid profile is the type of diet. A high supply of exogenous cholesterol at an insufficient metabolic rate is responsible for cardiovascular incidents and increased mortality [12]. A diet containing saturated trans-fatty acid isomers, which are formed during the hydrogenation of vegetable oils to make more solid fats such as margarine, does not affect LDL levels but lowers HDL [26]. On the other hand, a vegetarian diet, independently of genetic factors, is associated with reduced HDL levels, although physical activity in this group may partially offset the adverse effect of this diet [27]. A similarly unfavorable lipid profile, i.e., significantly elevated TG and LDL levels, has been reported in subjects following a vegan diet [28].

Many previous epidemiological studies have demonstrated the beneficial effects of physical activity in the prevention of cardiovascular diseases. An increased risk of death from arteriosclerosis and heart attack is found in people who are not very physically active, which is usually accompanied by an unfavorable lipid profile and overweight [29,30,31,32,33,34,35,36,37]. For this reason, it is recommended in highly developed countries to engage in recreational activity and amateur sports according to the idea of health-related fitness (H-RF). In specific cases, there is a need to optimally select exercises and adjust the intensity and frequency of training sessions to the age, gender, and health status of the groups. The exercise tolerance of non-athletes varies greatly and must be considered for the safety of participants in training programs throughout the cycle.

Furthermore, it must be emphasized that the success of the recommended training program depends on the degree of motivation in the different exercising groups, and this, in turn, depends on the different hopes and expectations of the training, and the obstacles to its continuation. Analysis of different motives for engaging in high- or moderate-intensity training such as improving health, good looks, or physical fitness showed no relationship with the weekly total time spent on physical activity [38]. The reason for this was probably the large interindividual variation in the study population by sex and age (15–55). In the case of a more homogeneous group, it is easier to choose the right set of exercises. For men approaching middle age willing to improve both health and fitness, circuit training is a more appropriate choice [39,40] to meet health-related fitness criteria [41]. Despite numerous studies on the effects of different types of training on lipid profiles, the response to a specific training program oriented simultaneously to improve physical fitness and lipid status in men in a narrow age range is unknown.

The aim of the study was to determine the effect of the author’s training program based on functional and strength exercises, performed in the form of circuit training, on triglyceride levels, cholesterol fractions in men aged 35–40.

## 2. Materials and Methods

### 2.1. Study Group

The experiment conducted in this study involved a group of men aged 35–40 years (*n* = 30) selected by purposive sampling, with the inclusion criterion being the level of engagement in recreational physical activity determined before the experiment. Due to the high intensity of the training, individuals who had been doing recreational training supervised by a coach for at least 2 years were qualified to participate in the study. The participants were healthy, did not smoke cigarettes, and did not take any regular medication. Endocrine and circulatory abnormalities (ECG examination) were used as exclusion criteria. Detailed characterization of participants is presented in Table 1.

Based on individual values and BMI ranges (18.5–24.9, 25–29.9, 30–34.9), there were four people in the experimental group with normal weight, eight were overweight, and three were with class I or greater obesity. In the control group, there were five normal-weight people, eight were overweight, and two were with class I or greater obesity. 

According to the assumptions of conducting the pedagogical experiment, the intervention of the researchers concerned the manipulation of the lifestyles of the participants by introducing an experimental training program into their routine physical activity (Table 2). To conduct the experiment, the technique of working with two equal groups formed from previously qualified participants was used. The men were randomized into two groups: a control group and a study group (each consisting of 15 participants).

### 2.2. Research Program and Methodology

The first test for both subgroups was performed before the experiment. In the control group, participants continued their previous form of activity, and, between the first and the final examinations, they followed their previous program of recreational physical activity. In the study (experimental) group, a special training modification was introduced (independent variable), consisting in performing a 60-min personal endurance and strength training. The experiment lasted eight weeks. All training sessions were conducted in the afternoon during the spring before the onset of the COVID-19 pandemic. After its completion, both subgroups were subjected to a final examination (effect control) (Figure 1).

The dependent variables were TC, TG, and HDL levels determined in fasting venous blood collected in the morning (8:00 am), whereas LDL levels were calculated according to the formula: LDL = TC-HDL-C-TG-C/5 [17]. In the case of the evaluation of non-HDL levels, the following formula was used: non-HDL-C = TC-HDL-C [42]. The analytical determination of all lipid levels was performed at ALAB laboratories (laboratory diagnostician No. PWZDL 11614) using the spectrophotometric method/AU 680 in duplicates. The relative between-assay variability in triglycerides and cholesterol fractions was expressed in CV% and did not exceed 2.5%. 

Furthermore, each participant was instructed not to use specialized diets and supplementation during the experiment due to the fact of the high relationship between diets and cholesterol levels. The participants in both groups did not change their diets during the experiment. Diets were monitored using notebooks in which the participants recorded the foods they consumed using home measures based on a photo album of foods and products provided. The recording procedure was continued for 3 days: 2 working days, 1 day off.

Analysis of diet observations revealed neither specific diets nor the use of performance-enhancing supplements in the training groups.

Participants were informed of all research procedures prior to participation in the study in accordance with the ethical principles of the WMA Declaration of Helsinki (2000). Participation in the project was voluntary and all participants were familiarized with the assumptions of the experiment. The precondition for participation in the study was the participant’s written informed consent. None of the participants had any medical contraindications to exercise, which was confirmed by an earlier medical certificate of no contraindications to taking part in the present experiment. The experiment was approved by the Bioethics Committee at the Regional Medical Chamber (No. 309/KBL/OIL/2019).

### 2.3. Description of the Exercises Included in the Experimental Program

The experimental program consisted in performing endurance and strength training. With its high diversity, the author’s program was supplemented with new methods and exercise structures currently considered to be the most effective. The main part of the training was based on different variants of circuit training [19,20,21] and the principles of functional training [22] (Table 2).

The concept of the experimental training program was based on the principle of small strength circuits [23] in which an additional assumption is to perform appropriately grouped exercises (strength, general fitness, and functional exercises). Each training unit is divided into three small circuits (Table 3):strength conditioning circuit (using barbells, dumbbells, and kettlebells);general fitness circuit (cardio exercises such as plyometrics and coordination exercises);functional fitness circuit (exercises involving muscle parts most commonly used in activities of daily living).

The performance of each exercise was limited by a time of 30 s, with the exception of the strength conditioning circuit, the goal of which was to complete 15 repetitions within 45 s. The training session began with warm-up exercises (5–10 min). Each training unit ended with stretching (10–15 min).

The weights of the kettlebells, dumbbells, and barbells were adjusted to the weight-dependent strength of the participant and did not change over the course of the training program.

In order to avoid fatigue and training monotony, an innovative training unit consistent with the entire program was planned for each session of the week. Each training session was preceded by a warm-up and included well-thought-out exercises according to the author’s idea. The training program was designed so that it was simple to perform and accessible to every participant. Individual exercises were performed at a fast pace (concentric phase of 1 s, eccentric phase of 2 s), with a particular focus on the correct technique. The study group trained three times a week, together under the supervision of the coach using the appropriate intensity and number of repetitions. Before the experiment, the participants underwent instruction and learned how to correctly perform the exercises. 

### 2.4. Statistical Analysis

Statistical data were compiled using Statistica software (version 13.3, Kraków, Poland). The Shapiro–Wilk test was used to test the data for normal distribution. Basic descriptive statistics (arithmetic mean, median, minimum, maximum, and standard deviation) were computed. Hypotheses were verified using the Student’s *t*-test for independent variables in the case of between-group comparisons and dependent variables when comparing pre- and post-test values across groups. Power of the test was expressed using Cohen’s d value. Statistical significance was set at *p* < 0.05.

## 3. Results

Table 4, Table 5, Table 6, Table 7 and Table 8 show the means, standard deviations, medians, concentration ranges, and results of the statistical analysis of the pre-test and post-test lipid profiles for each subgroup. 

Table 4 shows that there were statistically significant changes in mean serum total cholesterol levels in both groups after the experiment. In the experimental group, the time-dependent changes in the concentrations were four times greater than in the control group, whereas baseline TC-C levels were similar.

The results of HDL determinations contained in Table 5 revealed no significant post-test changes in this fraction in the experimental and control groups. There were also no between-group differences on the two test dates.

The baseline TG level was over 15% higher in the experimental group, but the difference was not statistically significant. A significant decrease (by 23%) was found in the experimental group for the TG fraction. Between-group differences were demonstrated for post-test changes in TG (4.1%).

Table 7 shows a significant decrease (by almost 30%) in LDL levels in the experimental group and an insignificant decrease (by 6%) in the fraction in the control group. Consequently, a significant, more than a fourfold between-group change in LDL levels was found at the end of the experiment.

The results presented in Table 8 show the changes in mean serum non-HDL cholesterol levels. There was a significant post-test decrease in the fraction (by 26%) in the experimental group and a much smaller decrease (by 4.1%) in the control group. There were significant between-group changes induced by the training. They are very similar to those for LDL because HDL levels were virtually unchanged and similar in both groups.

The reference range for TC determined in the laboratory is <190 mg/dL. In both the experimental and control groups, its mean levels were above normal. At the end of the 8-week experiment, there was a statistically significant decrease (by 19.4%) in total cholesterol levels in the experimental group (Ex) that performed the recommended training, with the mean below the recommended reference value. In the control group (Co), the decrease was much smaller, and its mean values failed to reach those recommended by the laboratory. The post-test between-group difference in mean cholesterol levels and the absolute mean post-test changes (ΔT) showed a statistically significant between-group difference. Mean HDL levels did not change significantly in both groups after the experiment. However, they decreased less in the Experimental group, and the absolute mean post-training changes (ΔT) showed a statistically significant between-group difference. Mean serum HDL levels were normal both pre- and post-test in both groups (recommended >40 mg/dL). The reference range for TG is <150 mg/dL. Absolute mean post-training changes (ΔT) showed a statistically significant between-group difference.

## 4. Discussion

The results of various studies on the effects of different types of training on lipid profiles often led to contradictory conclusions. The reasons for these discrepancies are the different experimental methodologies i.e., experimental groups of different ages, differences in baseline physical fitness, different training programs, and their duration, insufficient dietary control, and the season of the year and its significant effect on lipid profiles omitted from the experimental protocols [43]. Therefore, it is difficult to compare the results of different studies and find general relationships between the type of training and the final biochemical and biophysical effects. The 8-week circuit training used in our study had a beneficial effect on TC, TG, non-HDL, and LDL, but did not change HDL levels, the fraction that lowers the risk of coronary heart disease [44,45,46]. The findings of many studies on the effects of single training sessions may suggest that HDL-C is less affected than other lipids after a single exercise or a training period. A single exercise at an intensity equivalent to 90% of VO_2_max performed to exhaustion resulted in a decrease in LDL and HDL levels by 45% and 15%, respectively, during short-term recovery [47], which means that the proportion between these fractions changed in a way that is beneficial to health. Changes in HDL following a resistance training session occurred only for a low-intensity, high-repetition protocol [48]. A 1-h effort at an intensity of 60% VO_2_max resulted in a significant reduction in TC, TG, and LDL levels during the 72-h recovery period but a much smaller relative and transient change in HDL [49].

The relative HDL responses to the training period also appear to be much smaller than the changes in the other lipids. Several studies indicate any changes of HDL in response to a training. For this reason, and because of the role of this fraction in protecting against the development of arteriosclerosis and other cardiovascular complications, research should focus on the identification of the type of physical training that induces even a small increase in this cholesterol fraction. Because most of the studies on the effect of training suggest post-training decreases in LDL, TG, and TC levels, and few researchers have reported an increase in HDL levels, therefore in the discussion of the results of the present study we attempted to provide the explanation for the lack of post-training changes in this fraction. To this end, we analyzed the changes in lipid levels dependent on different types of training presented by other authors.

The variety of exercises used by different researchers requires a major simplification in the classification of training sessions by exercise intensity and volume. Training based on aerobic endurance exercise is characterized by a low to moderate intensity per session but a significant energy expenditure. On the other hand, training with predominantly anaerobic and strength exercises requires more intensive work. High-intensity interval training (HIIT) falls into this category. Furthermore, resistance training can be performed using high intensity/low repetitions sessions or low intensity/high repetitions sessions.

An example of using HIIT training to improve HDL-C is the adaptations induced by repeated 4 × 800 runs at 90% of HRmax. Three sessions of such training per week performed for 8 weeks resulted in an 18.1% increase in HDL, although TC was not altered [50]. Low-intensity aerobic training (50% VO_2_max) performed for 5 months significantly increased HDL by an average of 9.8%, with individual changes positively correlating (r = 0.75) with weekly time spent on exercising [51]. A similar increase in HDL (by 7% on average) was dependent on jogging distance (7–14 miles/week) [52]. A very large (19%) increase in this fraction was observed in young judo athletes who performed 18-min runs at an intensity of 65% VO_2_max for 8 weeks [53]. Small changes in men (by 3%) and slightly larger changes in women (by 4.3%) were observed after 20 weeks of aerobic training (3 sessions of 30 min/week) with an initially low intensity that was increasing in the following weeks [54]. Some studies suggest that HDL does not change after training in obese or overweight men [55,56,57,58], while after a 12-week training with moderate intensity and volume, overweight women (BMI ca. 30 kg/m^2^) reported a 24% increase in HDL levels, with no change in body weight [59]. A review article [60] quoted two original papers suggesting that, unlike resistance training, aerobic exercise improves HDL status to a greater extent but only at a sufficiently high intensity. Unfortunately, it is difficult to compare intensity across the papers cited because some researchers have characterized the intensity of training sessions based on VO_2_ values, while others used HR. Furthermore, the authors found a huge scatter of post-training adaptations for both resistance and aerobic training, which may in part be due to the different durations of the training period. The findings presented in studies on the effects of resistance training on HDL levels are inconclusive. Some authors have shown that the cumulative effect of resistance training induced only a 1% increase whereas aerobic training led to a 4% increase in HDL levels [61], which could demonstrate the major contribution of endurance exercises in improving this fraction. However, other studies have demonstrated that resistance training can also contribute to significant increases in HDL, as evidenced by a more than 11% increase in fraction levels following 21 weeks of very high-intensity resistance training [62]. An analysis of multiple factors affecting HDL was performed for a large population of Taiwanese residents. It was found that both aerobic and resistance training protocols were associated with higher HDL levels in athletes compared to the non-athlete group, but the magnitude of this parameter depended on multiple variables characterizing the participants [63]. A significant and independent factor affecting HDL levels by type of training was age. In the group of individuals aged 30 to 40 years, higher HDL was associated with resistance training compared with aerobic training and inactivity. In contrast, in older adults (>51 years), higher HDL levels were noted in the group performing aerobic exercises. Furthermore, with age, the difference in HDL-C between active and inactive individuals became blurred, and with increasing BMI values, HDL gradually decreased regardless of the type of physical activity. Smoking had a similar negative effect on HDL-C status.

Based on the studies cited, it can be expected that the endurance exercise used in our study and general weekly activity was insufficient to increase HDL levels in this group of men. 

On the other hand, in overweight and obese people, higher physical loads could be less tolerated before they became better adapted to exercise. However, the training program we used resulted in favorable changes in lipid profiles both in participants with elevated baseline TC, LDL, and TG levels and in those who had TC levels (162, 184, and 146 mg/dL) within the reference range. However, greater relative reductions in TC, LDL, and TG were observed in the other 12 participants with moderate or greater hyperlipidemia. This is consistent with the observations of other authors, who reported significantly higher baseline lipid levels in overweight or obese men [64]. This fact indicates a substantially higher proportion of post-training adaptations in hyperlipidemic people in the entire heterogeneous group of intervention participants.

It is difficult to interpret the small (by 4.7%) yet statistically significant reduction in TC levels following the study in the control group. It is possible that, being aware of the impact of lipid profiles on health, people in this group slightly changed their lifestyles during the study period. It should be noted that regular increased physical activity results not only in the normalization of the lipid profile but also in the improvement of the overall psychophysical state referred to as well-being. The metabolic and physical effects obtained after a relatively short training program were consistent with the expectations of the participants in the experimental group, who favored the improvements in physical fitness more than those concerning lipid profiles. Both goals were achieved, and at the end of the training, all exercisers expressed satisfaction from participating in the experiment and unanimously found the training sessions enjoyable. This fact may determine the successful promotion and continuation of this form of activity in men aged 35–40 years. This presumption is due to the fact that high levels of perceived enjoyment are more easily achieved during high-intensity circuit training than during aerobic moderate-intensity training [65,66]. 

In our study, the experimental group is too small to be divided into two homogeneous subgroups, namely one with ‘normal’ and the other with elevated lipids. Nevertheless, it seems advisable to analyse the changes depending on the baseline values. In addition, it seems appropriate to define a criterion for the qualification of subjects into groups of more or less responders.

It can be expected that the training program used in the study meets the criteria of the health-related fitness concept and can be promoted in this population as a strategy to improve health [67,68].

### Limitation of the Study

Unfortunately, the authors do not know whether the program’s participants permanently changed their lifestyles or how long the changes in their lipid profiles persisted after they stopped exercising.

## 5. Conclusions

The use of 8-week endurance and strength exercises in the form of circuit training leads to improvements in lipid profiles in men aged 35 to 40 years. 

### Practical Implication

The proposed training program can be used to support the treatment of excessively high cholesterol levels in order to lower them or change the LDL/HDL ratio. 

## Figures and Tables

**Figure 1 jcm-11-05146-f001:**
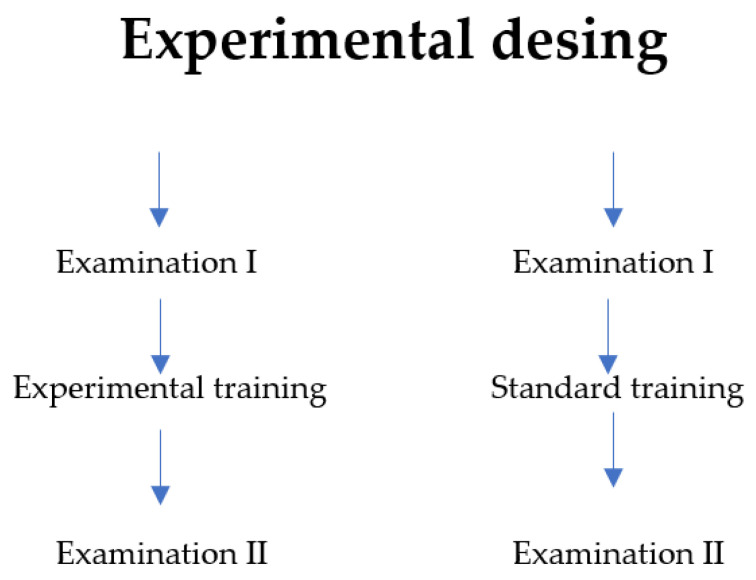
Training scheme. Source: own study.

**Table 1 jcm-11-05146-t001:** Characteristics of study participants in experimental and control groups.

Group	Variable	Mean	SD	Min–Max
Experimental N = 15	Age (years)	36.7	1.7	35–40
	Body height (cm)	178.3	4.9	170–189
	BM (kg)	91.24	13.1	62.3–112.6
	BMI (kg/m^2^)	28.2	4.1	23.2–36.4
Control N = 15	Age (years)	38.4	1.8	35–40
	Body height (cm)	177.8	7.4	163–192
	BM (kg)	87.8	14.4	65.4–121.2
	BMI (kg/m^2^)	26.9	2.9	21.2–31.6

Abbreviations: BM—Body Mass, BMI—Body Mass Index.

**Table 2 jcm-11-05146-t002:** Experimental training program: methodical assumptions.

Experimental Program Based on the Circuit Training	
Number of circuits	3 circuits × 3 type of training units
Number of exercises in a circuit	5
Number of repetitions or duration of the set	1 circuit–15 repetitions in 45 s 2–3 circuits–exercises for 30 s
Percentage of maximum weight	50% (individual levels)
Exercise intensity	Fast
Rests between circuits	1 min between exercises–3 min between circuits

**Table 3 jcm-11-05146-t003:** Training program used in the research project.

Strength Conditioning Circuit	General Fitness Circuit	Functional Fitness Circuit
Overhead squat	Footwork using speed ladders	Chest press with rubbers attached to the ladder
Narrow grip chest pull-ups	Transition from standing position to the push-up position followed by a tuck jump (burpees)	Lunges with moving the medicine ball from over the head
Lying alternate dumbbell press on the exercise ball	A-skips with dumbbells held in hands	Push-ups on Bosu ball and return to the standing position while alternate lifting the Bosu ball over the head and chest press
Kettlebell swing in from the sumo position	Jumping over the hurdle combined with moving under the hurdle	Pull-ups on the hanging rope with trunk rotations
Russian twists with the disc weight held with both hands	Squats followed by single leg jumps on a plyo-box	Side stepping between the discs

**Table 4 jcm-11-05146-t004:** TC-C level (mg/dL).

TC	Experimental Group					Control Group					Between Groups Differences		
	Mean	Med	Min	Max	SD	Mean	Med	Min	Max	SD	t_1_	*p*	d
Pre	213.3	209	146	301	41.3	209.3	216	145	265	35.3	0.28	0.781	0.10
Post	171.9	173	122	250	31.5	199.5	200	132	247	34.9	−2.27	**0.031**	0.83
Pre-post differences	41.3	36	24	51		9.74	16	13	18	-	-	**0.001**	
Cohen’s d	**t_2_ = 5.7 *p* < 0.001** **1.47**					**t_2_ = 2.39 *p* = 0.031** **0.57**							

TC—Total Cholesterol; Med—Median; Min—minimum; Max—maximum; SD—Standard Deviation; t_1_—Student’s *t*-test for independent variables; t_2_—Student’s *t*-test for dependent variables; d—effect size; Values in bold that are statistically significant.

**Table 5 jcm-11-05146-t005:** Serum HDL level (mg/dL).

HDL	Experimental Group					Control Group					Between Groups Differences		
	Mean	Med	Min	Max	SD	Mean	Med	Min	Max	SD	t_1_	*p*	d
Pre	51.9	49	37	71	10.0	53.7	51	32	82	12.5	−0.45	0.667	0.19
Post	49.9	45	35	70	11.1	50.5	49	25	73	14.0	−0.12	0.909	0.08
Pre-post differences	2	4	2	1		3.2	2	7	9			0.735	
Cohen’s d	t_2_ = 0.98 *p* = 0.344 0.253					t_2_ = 1.06 *p* = 0.308 0.268							

HDL—High Density Lipoprotein; Med—Median; Min—minimum; Max—maximum; SD—Standard Deviation; t_1_—Student’s *t*-test for independent variables; t_2_—Student’s *t*-test for dependent variables; d—effect size.

**Table 6 jcm-11-05146-t006:** Serum TG level (mg/dL).

TG	Experimental Group					Control Group					Between Groups Differences		
	Mean	Med	Min	Max	SD	Mean	Med	Min	Max	SD	t_1_	*p*	d
Pre	142.2	132	69	238	65.4	119.7	117	57	185	40.0	1.14	0.264	0.41
Post	108.6	90	44	254	42.5	125.1	119	62	241	50.7	−0.85	0.404	0.31
Pre-post differences	33.6	42	25	16		−5.47	−2	−5	−56	-	-	**<0.001**	
Cohen’s d	**t_2_ = 3.07 *p* = 0.008** **0.788**					t_2_ = −0.49 *p* = 0.631 0.115							

TG—Triglycerides; Med—Median; Min—minimum; Max—maximum; SD—Standard Deviation; t_1_—Student’s *t*-test for independent variables; t_2_—Student’s *t*-test for dependent variables; d—effect size; Values in bold that are statistically significant.

**Table 7 jcm-11-05146-t007:** Serum LDL level (mg/dL).

LDL	Experimental Group					Control Group					Between Groups Differences		
	Mean	Med	Min	Max	SD	Mean	Med	Min	Max	SD	t_1_	*p*	d
Pre	132.9	125	78	228	42.5	131.6	130	45	190	38.6	0.09	0.931	0.03
Post	99.1	101	49	156	29.4	124.0	137	45	176	37.6	−1.92	0.064	0.74
Pre-post differences	33.8	24	29	72	-	7.6	-	0	35	-	-	**<0.001**	
Cohen’s d	**t_2_ = 4.25 *p* = 0.001** **1.18**					t_2_ = 1.69 *p* = 0.113 0.469							

LDL—Low Density Lipoprotein; Med—Median; Min—minimum; Max—maximum; SD—Standard Deviation; t_1_—Student’s *t*-test for independent variables; t_2_—Student’s *t*-test for dependent variables; d—effect size; Values in bold that are statistically significant.

**Table 8 jcm-11-05146-t008:** Non-HDL cholesterol level (mg/dL).

non-HDL	Experimental Group					Control Group					Between Groups Differences		
	Mean	Med	Min	Max	SD	Mean	Med	Min	Max	SD	t_1_	*p*	d
Pre	161.4	157	115	261	46.9	155.5	157	63	219	43.9	0.37	0.714	0.12
Post	122.2	119	79	207	32.7	149.1	154	68	200	40.5	1.96	0.061	0.72
Pre-post differences	42.2	38	36	54	-	6.4	12	23	42	-	-	**<0.001**	
Cohen’s d	**t_2_ = 5.3 *p* < 0.001** **1.344**					t_2_ = −1.35 *p* = 0.197 0.376							

non-HDL—non High Density Lipoprotein; Med—Median; Min—minimum; Max—maximum; SD—Standard Deviation; t_1_—Student’s *t*-test for independent variables; t_2_—Student’s *t*-test for dependent variables; d—effect size; Values in bold that are statistically significant.

## Data Availability

The data presented in this study are available on request from the corresponding author.

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
