# Peer review of "Effect of High-Intensity Strength and Endurance Training in the Form of Small Circuits on Changes in Lipid Levels in Men Aged 35–40 Years"

_jcm, 2022, doi:10.3390/jcm11175146_

Round 1
Reviewer 1 Report
In this study the authors examine the effect of high-intensity strength and endurance training in the 2 forms of small circuits on changes in lipid levels in men aged 3 35-40 years.
Although the study has the potentiality of being shared with the scientific community, I believe that the manuscript would benefit from a minor revision with the attempt to better support their experimental setting.
1. More information should be provided about the Experimental
2. The intervention protocol should be better described.
3. The Discussion should be enriched with the existing theory. The authors should clearly describe the scientific evidence that supports their findings.
4. I would like to see more of the practical implications. Based on the analyzed variables, how the authors intend to use their findings?
Kind regards
Author Response
Dear Reviewer,
Thank you very much for your time and valuable comments, which all have been considered and incorporated. The detailed list of responses is given below. We hope that the modifications and explanation will be acceptable for you.
Yours sincerely,
Rydzik, corresponding author
In this study the authors examine the effect of high-intensity strength and endurance training in the 2 forms of small circuits on changes in lipid levels in men aged 3 35-40 years.
Although the study has the potentiality of being shared with the scientific community, I believe that the manuscript would benefit from a minor revision with the attempt to better support their experimental setting.
1.More information should be provided about the Experimental W.
A: In the manuscript, based on individual BMI values, the subjects' weight was classified into three categories; normal, overweight and obese. The data are included under Table 2 . In addition, a graphical diagram was added and a description was completed
2.The intervention protocol should be better described.
A: A qualitative description of the entire training programme, which consisted of three types of circuit exercises, can be found in Table 3. The imposed interval time and the work, while the length of the breaks between circuits was variable (1-3 min) and depended on the fitness of the exercisers. In the case of strength exercises, kettlebell barbell weights and dumbbells, the disk at the beginning of the programme was individually adjusted to the body weight and strength of the exerciser and did not change until the end of the training period. We tried to improve the description of the intervention
3.The Discussion should be enriched with the existing theory. The authors should clearly describe the scientific evidence that supports their findings.
A:The literature cited in the discussion and our own results support the theory of a cause-effect relationship between moderate physical activity and favourable health status. In health sciences, this theory is known as health-related fitness. Supported the scientific evidence with literature.
4.I would like to see more of the practical implications. Based on the analyzed variables, how the authors intend to use their findings?
A: Based on the data obtained, a training programme is being prepared for middle-aged, inactive and moderately overweight men as a safe and health-promoting way to improve physical fitness and lipid profile. In addition, an application form has been added.
Reviewer 2 Report
The authors carry out another study to assess the effect of a physical exercise program on plasma lipid levels. They have in their favor that they study a homogeneous population in terms of sex, age and amount of initial physical activity. But this population is heterogeneous in terms of previous values of the lipid profile. The work is interesting but the information is insufficient.
The authors only write about the mean values of each group and their changes. In lines 272-73, they indicate that the normal reference value for TC is 190 mg/dl. I ask: does the decrease in TC values occur in subjects with hypercholesterolemia, in normal subjects or in all? It would be interesting to know to consider whether the program can have a therapeutic action or only a preventive one?
If the values only go down in “normal” subjects, the program would not be effective, even if it produces significant changes. Furthermore, whether the exercise program lowers TC, TG, or LDL values in subjects with "normal" values is of little clinical interest. The same as if it increases HDL values in subjects with “high or normal” values.
As we can see from the minimum and maximum values of each variable, each of the groups is made up of a mixture of subjects. There are people with "normal" lipid values according to laboratory references and others with "high" values. I don't think it's interesting to see what happens in a “mix” group. What is really interesting is to know if the exercise program is useful for bringing subjects with “abnormal” values back to normal and/or keep normal subjects normal. I think what is interesting is that the population has "normal" values, not "low or high" values.
Regarding HDL cholesterol. If the authors' exercise program increases values in subjects with low baseline values and maintains "normal" values in "normal" subjects, the program will be successful, but those changes may not be statistically significant, since the group is heterogeneous and the answer, possibly, too. The most useful research question would be: Does the training program normalize HDL values in subjects with low values? If this were so, the program would be successful, regardless of its action on HDL in normal subjects. More normal than normal is of no interest.
In the work, it is observed that TC, TG and LDL decrease significantly, it can be assumed that this is due to the decrease in values in subjects with "high" values initially, but it cannot be guaranteed.
To avoid this, authors should subdivide groups based on “normal” baseline values or presence/absence of cardiovascular risk and not use a heterogeneous or “mixed” group. They could also write for each variable what has happened in the “normal” subjects and in the “cardiovascular risk” subjects.
Preliminary, I raise the following doubts and points for improvement for the work, leaving other aspects for a possible second round.
1.- Table 1 does not indicate how many subjects are part of the experimental and control groups.
They use only the acronym BM to indicate "Body mass" and "body height" without abbreviations. “BMI is not described at the foot of the table or in the text.
2.- The authors do not indicate how the system for assigning the subjects to each group was.
3.- The values in table 2 are not clear. There are apparent contradictions with the text. What does number of circuits 3x3 mean?
4.- Tables 4 to 9 the name of the row "Between group differences" does not seem correct. The differences that this row shows are not between groups, they are between pre-test and post-test.
5.- The authors do not indicate if all the subjects of the research group trained together, or if they formed different groups, or did it individually.
6.- The authors do not report on the instruction and supervision of the exercises.
7 .- The authors could create their own subsection for the statistical procedure.
8.- Lines 193 and 194. The authors repeat “Statistical significance for differences was calculated using the student’s t-test for independent samples”.
9.- In the statistical procedure it is not written what it is used for: "Cohen's d"
10.- In the statistical procedure the authors indicate that they use the coefficient of variation (VC), but it is not in any result.
11.- About the discussion.
Most of the discussion paragraphs are summaries of other studies. The authors do not discuss their results and do not answer the following questions that would be interesting to raise.
a.- Are you interested in lowering lipids and/or increasing HDL in people with normal values?
b.- Does the effect of the exercise program on normal subjects mask the result?
c.- If the subjects were already used to intense physical exercise. Why are there subjects with “very high” values of LDL, TC and TG? Have these subjects responded to the new program?
d.- Is there a relationship between the response to the exercise program and the diet of each subject? Have those with a healthier diet responded better?
e.- Why does TC decrease significantly in the control group? How does this fact influence the interpretation of the results of the experimental group?
12.- In the discussion, the authors do not show the limitations of their study.
13.- On the conclusions.
The first conclusion cannot be accepted because the number of subjects with cardiovascular risk at baseline is not known. If the changes have occurred in "normal" subjects, the risk remains the same, even if there are significant differences.
The second conclusion is not related to the objectives and there are no results that support it, it is an opinion of the authors and its place is in the discussion.
Author Response
Dear Reviewer,
Thank you very much for your time and valuable comments, which all have been considered and incorporated. The detailed list of responses is given below. We hope that the modifications and explanation will be acceptable for you.
Yours sincerely,
Rydzik, corresponding author
Reviewer 2
The authors carry out another study to assess the effect of a physical exercise program on plasma lipid levels. They have in their favor that they study a homogeneous population in terms of sex, age and amount of initial physical activity. But this population is heterogeneous in terms of previous values of the lipid profile. The work is interesting but the information is insufficient.
The authors only write about the mean values of each group and their changes. In lines 272-73, they indicate that the normal reference value for TC is 190 mg/dl. I ask: does the decrease in TC values occur in subjects with hypercholesterolemia, in normal subjects or in all? It would be interesting to know to consider whether the program can have a therapeutic action or only a preventive one?
A: The upper physiological TC concentration was set at 190 mg/dl. All 15 subjects showed a decrease in TC after training. Before training, 3 subjects presented concentrations lower than 190 mg/dl (162, 184, 146 mg/dl) and after training concentrations decreased by 4,21 and 24 mg/dl, respectively. In the remaining 12 subjects, baseline TCs were ≥ 190 mg/dl and after training only 3 subjects remained relatively hypercholestrolemic. The relative post-training TC reduction correlated (r=-0.46) with baseline levels. This means that the mean post-training TC decrease was 21% responsible for the baseline concentration. In other words, greater absolute and relative post-training TC changes are recorded in baseline hypercholesterolaemia than in 'normal' individuals
If the values only go down in “normal” subjects, the program would not be effective, even if it produces significant changes. Furthermore, whether the exercise program lowers TC, TG, or LDL values in subjects with "normal" values is of little clinical interest. The same as if it increases HDL values in subjects with “high or normal” values.
As we can see from the minimum and maximum values of each variable, each of the groups is made up of a mixture of subjects. There are people with "normal" lipid values according to laboratory references and others with "high" values. I don't think it's interesting to see what happens in a “mix” group. What is really interesting is to know if the exercise program is useful for bringing subjects with “abnormal” values back to normal and/or keep normal subjects normal. I think what is interesting is that the population has "normal" values, not "low or high" values.
Regarding HDL cholesterol. If the authors' exercise program increases values in subjects with low baseline values and maintains "normal" values in "normal" subjects, the program will be successful, but those changes may not be statistically significant, since the group is heterogeneous and the answer, possibly, too. The most useful research question would be: Does the training program normalize HDL values in subjects with low values? If this were so, the program would be successful, regardless of its action on HDL in normal subjects. More normal than normal is of no interest.
A: In most cases reported in the literature, it has been shown that HDL-C hardly undergoes the expected changes in response to training, regardless of its baseline concentration. Despite this, training is also considered to have a beneficial effect on the lipid profile if triglyceride and/or LDL concentrations are lowered with unchanged HDL-C. A quantitative measure of the effectiveness of a training programme is the reduction in risk of cardiovascular incidents as estimated by the Atherogenic Index of Plasma (AIP=log (TG/HDL-C) (Sadeghi M 2021 in Advances in Medical Sciences)
In the work, it is observed that TC, TG and LDL decrease significantly, it can be assumed that this is due to the decrease in values in subjects with "high" values initially, but it cannot be guaranteed.
A: This is true. The post-training relative decreases in TC, TG and LDL have been shown to be twice as large for high concentrations of the aforementioned lipids, which is typical for overweight/obese individuals (Ouerghi N., et al. 2017 Biology of Sport ) In our study, the experimental group is too small to be divided into two homogeneous subgroups one with 'normal' and the other with elevated lipids. Nevertheless, it seems advisable to analyse the changes depending on the baseline values. In addition, it seems appropriate to define a criterion for the qualification of subjects into groups of "more- or less responders".
To avoid this, authors should subdivide groups based on “normal” baseline values or presence/absence of cardiovascular risk and not use a heterogeneous or “mixed” group. They could also write for each variable what has happened in the “normal” subjects and in the “cardiovascular risk” subjects.
A: The reviewer's suggestion is pertinent but will require research to be undertaken on a larger group
Preliminary, I raise the following doubts and points for improvement for the work, leaving other aspects for a possible second round.
1.- Table 1 does not indicate how many subjects are part of the experimental and control groups.
A: This has been corrected
They use only the acronym BM to indicate "Body mass" and "body height" without abbreviations. “BMI is not described at the foot of the table or in the text.
Uzupełnione w Tabeli 1
2.- The authors do not indicate how the system for assigning the subjects to each group was.
A: In the manuscript it was stated that experience of physical activity at a recreational level was a prerequisite for participation. We have added a diagram showing the study design. In contrast, a description of randomisation is included in the methodology
3.- The values in table 2 are not clear. There are apparent contradictions with the text. What does number of circuits 3x3 mean?
A: This has been corrected. 3 qualitative circuit types and 3 circuits in for each type.
4.- Tables 4 to 9 the name of the row "Between group differences" does not seem correct. The differences that this row shows are not between groups, they are between pre-test and post-test.
A: This has been corrected pre-post differences
5.- The authors do not indicate if all the subjects of the research group trained together, or if they formed different groups, or did it individually. they trained together
A: The experimental group trained together under the supervision of an instructor. Added information in the text
6.- The authors do not report on the instruction and supervision of the exercises.
A: Added information in the text
7 .- The authors could create their own subsection for the statistical procedure.
A: This has been corrected
8.- Lines 193 and 194. The authors repeat “Statistical significance for differences was calculated using the student’s t-test for independent samples”.
A: This has been corrected
9.- In the statistical procedure it is not written what it is used for: "Cohen's d"
A: This has been corrected - Cohen`d is index of test power
10.- In the statistical procedure the authors indicate that they use the coefficient of variation (VC), but it is not in any result.
A: This has been corrected
11.- About the discussion.
Most of the discussion paragraphs are summaries of other studies. The authors do not discuss their results and do not answer the following questions that would be interesting to raise.
a.- Are you interested in lowering lipids and/or increasing HDL in people with normal values?
b.- Does the effect of the exercise program on normal subjects mask the result?
c.- If the subjects were already used to intense physical exercise. Why are there subjects with “very high” values of LDL, TC and TG? Have these subjects responded to the new program?
d.- Is there a relationship between the response to the exercise program and the diet of each subject? Have those with a healthier diet responded better?
e.- Why does TC decrease significantly in the control group? How does this fact influence the interpretation of the results of the experimental group?
A: The discussion has been extended and modified
12.- In the discussion, the authors do not show the limitations of their study.
A: The main study limitation is small sample size. Hence it was impossible to create homogenous subgroups instead analyzing one mixed and heterogenic group. Added limitation of the study.
13.- On the conclusions.
The first conclusion cannot be accepted because the number of subjects with cardiovascular risk at baseline is not known. If the changes have occurred in "normal" subjects, the risk remains the same, even if there are significant differences.
The second conclusion is not related to the objectives and there are no results that support it, it is an opinion of the authors and its place is in the discussion.
A: This has been corrected
Round 2
Reviewer 2 Report
The authors should include in the discussion a paragraph justifying why they have not separated the population into two groups according to the normality of blood lipid values (normal/high).
In the bibliography:
1. They must correct the excess of capital letters, for example, in the numbers: 29, 31, 41, 46, among others.
2. Complete and modify, for example, references: 39, 40, 48.
3. Write the titles of the magazines in the same way: either all in abbreviations or all with full name.
4. Check errata, for example, in punctuation (47), bold, italics.
5. Review other possible typos
Author Response
Dear Reviewer,
Thank you very much for your time and valuable comments, which all have been considered and incorporated. The detailed list of responses is given below. We hope that the modifications and explanation will be acceptable for you.
Yours sincerely,
Rydzik, corresponding author
The authors should include in the discussion a paragraph justifying why they have not separated the population into two groups according to the normality of blood lipid values (normal/high).
A: Added in discussion
In the bibliography:
They must correct the excess of capital letters, for example, in the numbers: 29, 31, 41, 46, among others
A: This has been corrected
Complete and modify, for example, references: 39, 40, 48.
A: This has been corrected
Write the titles of the magazines in the same way: either all in abbreviations or all with full name.
A: This has been corrected
Check errata, for example, in punctuation (47), bold, italics.
A: This has been corrected
Review other possible typos
A: Literature was compiled using Mendeley using the journal template